# Research on Hyper-Parameter Optimization of Activity Recognition Algorithm Based on Improved Cuckoo Search

**DOI:** 10.3390/e24060845

**Published:** 2022-06-20

**Authors:** Yu Tong, Bo Yu

**Affiliations:** 1School of Computer Science and Technology, Hefei Normal University, Hefei 230601, China; 2School of Civil Engineering, Hefei University of Technology, Hefei 230009, China; yubochina@hfut.edu.cn

**Keywords:** activity recognition, cuckoo optimization algorithm, hyper-parameter

## Abstract

Activity recognition methods often include some hyper-parameters based on experience, which greatly affects their effectiveness in activity recognition. However, the existing hyper-parameter optimization algorithms are mostly for continuous hyper-parameters, and rarely for the optimization of integer hyper-parameters and mixed hyper-parameters. To solve the problem, this paper improved the traditional cuckoo algorithm. The improved algorithm can optimize not only continuous hyper-parameters, but also integer hyper-parameters and mixed hyper-parameters. This paper validated the proposed method with the hyper-parameters in Least Squares Support Vector Machine (LS-SVM) and Long-Short-Term Memory (LSTM), and compared the activity recognition effects before and after optimization on the smart home activity recognition data set. The results show that the improved cuckoo algorithm can effectively improve the performance of the model in activity recognition.

## 1. Introduction

In recent years, a number of methods have been proposed for activity recognition with non-obtrusive sensors [1,2], which have appeared as Ambient Intelligence (AmI) [3] and enablers to facilitate the development of applications that are aware of users’ presence and contexts, and are adaptive and responsive to their needs and habits. The methods mainly include conventional approaches and deep learning approaches, which have witnessed fast development and advancement in recent years [4]. Conventional approaches mainly use Decision Tree [5], K-Nearest Neighbor (KNN) [6], Support Vector Machine (SVM) [7], Naïve Bayes(NB) [8], Hidden Markov Model (HMM) [9], Conditional Random Field (CRF) [10] and other traditional machine learning methods [11]. Deep learning approaches mainly include Convolutional Neural Network (CNN) [12] and Recurrent Neural Network (RNN) [13]. Compared to conventional approaches, deep learning approaches could automatically extract appropriate features from raw sensor data during the training phase, and present the low-level original temporal features with high-level abstract sequences [14]. 

However, both conventional approaches, or traditional methods, and deep learning methods often contain some hyper-parameters which affect the performance of the model. Hyper-parameters are defined relative to general parameters. General parameters are variables obtained by model learning, while hyper-parameters are set variables, based on experience. Specifically, a typical traditional method is Support Vector Machine (SVM) [15,16], which has been proven to be sensitive to hyper-parameters C and λ, while a typical deep learning approach is CNN [17,18], which has been proven to be very sensitive to hyper-parameter network depth, the number of filters and their respective sizes. In addition, there are other methods that include hyper-parameters but are rarely optimized for activity recognition, such as Least Squares Support Vector Machine (LS-SVM) [19] and Long-Short-Term Memory (LSTM) [20]. 

Among these algorithms that contain hyper-parameters, some only contain continuous hyper-parameters, whose values are in a continuous interval, some only contain integer hyper-parameters, that can only take integer values, and some contain mixed hyper-parameters, containing both continuous hyper-parameters and integer hyper-parameters. Gradient algorithm is often used for continuous parameter optimization but is not suitable for hyper-parameter optimization. Thus, hyper-parameter optimization is important but faces challenging problems. To solve the hyper-parameter optimization problem, most researchers resort to intelligent evolutionary algorithms, which mainly include genetic algorithm [17,21] and particle swarm optimization (PSO) algorithm [18,22]. In recent years, Derivative-Free Optimization (DFO) was proposed by Koch et al. [23], and used to find optimal values for options of the global optimization solver BARON by Liu et al. [24]. Optuna [25] is also a hyper-parameter tuning algorithm that comes with Python and in recent years, it has been used to solve deep learning hyper-parameter optimization problems [26,27].

This article optimized the hyper-parameters in the activity recognition method based on the Cuckoo Search (CS). The CS [28] is an intelligent evolutionary algorithm with global convergence, proposed by Yang Xinshe, in 2009, to simulate the behavior of cuckoos selecting nests and laying eggs. However, the traditional cuckoo algorithm can only deal with continuous hyper-parameter optimization problems, and is powerless for integer hyper-parameter optimization problems and mixed hyper-parameter optimization problems. In order to solve integer hyper-parameter optimization and mixed hyper-parameter optimization, this paper improved the traditional cuckoo algorithm to adapt to different types of hyper-parameter optimization problems. Then, to verify the effectiveness of the algorithm, this paper used the algorithm to optimize the hyper-parameters in the LS-SVM and LSTM models which are commonly used for activity recognition.

The remainder of this paper is organized as follows. In Section 2, the traditional CS algorithm will be introduced and the improved CS algorithms will be given. In Section 3, this paper will introduce the related hyper-parameters in the LS-SVM and LSTM models. To illustrate the efficiency of the algorithm, in Section 4, simulation results are given for indoor positioning and activity recognition. Conclusions are drawn in Section 5.

## 2. Improved Cuckoo Optimization Search Algorithm

This section first introduces the traditional cuckoo optimization algorithm, and then gives two improved cuckoo algorithms.

### 2.1. Traditional Cuckoo Optimization Algorithm

The CS Algorithm assumes that the cuckoo’s behavior is in the following three ideal states: (1) The cuckoo lays only m eggs at a time, and randomly selects a suitable nest to hatch these m eggs. (2) In the process of cuckoo bird nest selection, the best quality bird nest will be retained for the next generation. (3) Under the premise of a certain number of bird nests that a cuckoo can choose from, the probability that each bird nest owner finds a foreign bird egg is *p*, where *p* ∈ [0,1]. If foreign bird eggs are found, the owner of the nest will rebuild a nest.

The flowchart of traditional the CS algorithm is presented in Figure 1.

The implementation of the traditional CS algorithm is as follows:

Step 1. Generate n random host nests $x_{i},i=1,\cdots ,n$, set *t* = 0 and compute the fitness $F_{i},i=1,\cdots ,n$.

Step 2. Find best_nest and best_fitness. If *t* < iter_num (the maximum number of iterations), go to Step 3, or else go to the last step.

Step 3. Update_nests: generate $x_{i}^{new},i=1,\cdots ,n$ with Lévy flights and compute the fitness $F_{i}^{new},i=1,\cdots ,n$, if $F_{i}^{new}>F_{i}$, update $x_{i}=x_{i}^{new}$.

Step 4. Abandon_nests: generate a random fraction P for every nest $x_{i},i=1,\cdots ,n$, if P < Pa, build a new one at new locations $x_{i}^{new}$ via Lévy flights, and update $x_{i}=x_{i}^{new}$.

Step 5. Calculate $F_{i}^{t},i=1,\cdots ,n$ at the nest $x_{i},i=1,\cdots ,n$ obtained in Step 4 in the *t*-th iterations, and find max_fitness $F_{}^{t\max}$ and the corresponding nest $x_{}^{t\max}$, if $F_{}^{t\max}>\mathrm{best}\_\mathrm{fitness}$, update best_nest = $x_{}^{t\max}$ and best_fitness = $F_{}^{t\max}$. 

Step 6. *t* = *t* + 1, if *t* < iter_num, and best_fitness < max, go to Step 3, or else go to Step 7.

Step 7. Return best_nest and best_fitness. 

### 2.2. Improved Cuckoo Optimization for Optimizing Integer Parameters

To optimize integer parameters, the host nests $x_{i},i=1,\cdots ,n$ need to be rounded. To this end, this article changed Steps 1, 3, and 4 in the cuckoo optimization algorithm to adapt to the optimization of integer parameters. In order to distinguish it from the basic cuckoo algorithm, this article refers to the above improved algorithm as cuckoo Algorithm 1.
**Algorithm 1** Improved Cuckoo Optimization for Optimizing Integer Parameters
**Step 1**. Generate *n* random host nests xi,i=1,⋯,n, rounding all nests, and compute the fitness Fi,i=1,⋯,n.**Step 2.** Find best_nest and best_fitness. If *t* < iter_num, go to Step 3, else go to the last step.**Step 3**. Update_nests: generate xinew,i=1,⋯,n with Lévy flights, rounding all nests, and compute the fitness Finew,i=1,⋯,n, if Finew>Fi, update xi=xinew.**Step 4**. Abandon_nests: generate a random fraction P for every nest xi,i=1,⋯,n, if fraction P < Pa, build a new one at new locations xinew via Lévy flights, rounding it, and update xi=xinew.**Step 5**. Calculate Fit,i=1,⋯,n at the nest xi,i=1,⋯,n in the *t*-th iterations, find max_fitness Ftmax and the corresponding nest xtmax, if Ftmax>best_fitness, update best_nest = xtmax and best_fitness = Ftmax.**Step 6.** *t* = *t* + 1, if *t* < iter_num, and best_fitness < max, go to Step 3, or else go to Step 7.**Step 7.** Return best_nest and best_fitness. 

### 2.3. Improved Cuckoo Optimization for Optimizing Continuous and Integer Mixed Parameters 

Assuming that the parameters to be trained include m continuous parameters and k integer parameters, the *i*-th host nest $x_{i}$ needs to include two parts: the continuous part $x_{i}^{m}$ and the integer part $x_{i}^{k}$. Therefore, the *i*-th host nest can be expressed as $x_{i}=\left[\begin{array}{cc} x_{i}^{m} & x_{i}^{k} \end{array}\right]$.

To optimize continuous and integer mixed parameters simultaneously, Steps 1, 3, and 4 in the cuckoo optimization algorithm need to be modified. To distinguish it from the above two previous cuckoo algorithms, this article refers to the above improved algorithm as cuckoo Algorithm 2.
**Algorithm****2** Improved Cuckoo Optimization for Optimizing Continuous and Integer Mixed Parameters**Step 1.** Generate *n* random host nests xi,i=1,⋯,n which include two parts, the random continuous part xim,i=1,⋯,n and the random integer part xik,i=1,⋯,n. Then, compute the fitness Fi,i=1,⋯,n.**Step 2.** Find best_nest and best_fitness. If *t* < iter_num, go to Step 3, else go to the last step.**Step 3.** Update_nests: generate host nests xinew,i=1,⋯,n with Lévy flights, rounding the integer part, and compute the fitness Finew,i=1,⋯,n, if Finew>Fi, update xi=xinew.**Step 4.** Abandon_nests: For every nest xi,i=1,⋯,n, generate a random fraction P, if fraction P < Pa, build a new one at new locations xinew via Lévy flights and rounding the integer part, then update xi=xinew.**Step 5.** Calculate Fit,i=1,⋯,n at the nest xi,i=1,⋯,n in the *t*-th iterations, find max_fitness Ftmax and the corresponding nest xtmax, if Ftmax>best_fitness, update best_nest = xtmax and best_fitness = Ftmax. **Step 6.**
*t* = *t* + 1, if *t* < iter_num, and best_fitness < max, go to Step 3, or else go to Step 7.**Step 7.** Return best_nest and best_fitness. 

## 3. Hyper-Parameters in LS-SVM and LSTM

### 3.1. Hyper-Parameters in LS-SVM

LS-SVM is an improvement on the standard SVM. It changes the insensitive loss function in SVM to a quadratic loss function, and replaces inequality constraints with equality constraints. The solving coefficient is greatly reduced, and the solving speed is accelerated.

LS-SVM is described as follows:

Take a given training dataset $\left\{x_{i},y_{i}\right\},i=1,\cdots ,n$, where $x_{i}$ is an n-dimensional input vector and $y_{i}$ is a one-dimensional output scalar. The optimization problem and the constraint conditions of LS-SVM algorithm are:(1)$$\underset{w,e}{\min}J(w,b,\xi )=\frac{1}{2}{\left\|w\right\|}^{2}+\frac{1}{2}\gamma \sum_{i=1}^{n}\xi_{i}^{2}$$
(2)$$s.t\;y_{i}\left[w^{T}\Phi \left(x_{i}\right)+b\right]=1-\xi_{i},\;i=1,\cdots ,n$$
where *w* ∈ *H* is the weight vector and H is the higher dimension space projected by the nonlinear function *φ*(*x*) from the original space *R*. Furthermore, $\xi_{i}\in R$ is the slack variable for $x_{i}$, which measures the deviation degree of a datum from the ideal condition of the classification model, and *b* ∈ *R* is the bias, *γ* is regularization factor. The regularization factor is similar to the penalty factor C in SVM and is used to adjust the confidence interval of LS-SVM and the proportion of empirical risk.

By solving the above optimization problem, the decision function for classification can be obtained [29] as: (3)$$y(x)=\mathrm{sgn}\left[\sum_{i=1}^{n}\alpha_{i}y_{i}K\left(x,x_{i}\right)+b\right]$$
where $K\left(x,x_{i}\right)$ is kernel functions, $\alpha_{i}$ are positive real constants and *b* is a real constant which can be obtained by solving the above optimization problem.

There are three commonly used kernel functions: linear kernel functions, polynomial kernel functions, and radial basis kernel functions. Among the three commonly used kernel functions, the radial basis kernel function can non-linearly map the samples to the high-dimensional space, and has fewer parameters and operations compared with the polynomial kernel function.

The width parameter σ (sigma) of the RBF function, together with the regularization parameter *γ* (gam), directly affect the performance of the LS-SVM model. However, the two hyper-parameters were continuously chosen as a fixed value according to experience. Therefore, it is very important to study the selection method of σ, and *γ*. In this paper, we used the cuckoo search algorithm to optimize the optimal gam and sigma, and then used the LS-SVM to identify user activities.

### 3.2. Hyper-Parameters in LSTM

LSTM is one of the RNNs and the primary objectives of LSTM are to model long-term dependencies and determine the optimal time lags for time series problems. These features are especially desirable for activity recognition, due to the lack of a priori knowledge on the relationship between prediction results and the length of input historical data. The LSTM architecture is composed of one input layer, one recurrent hidden layer, which has a basic unit that is memory block instead of traditional neuron node, and one output layer.

Suppose that the input sequence is denoted as $x=\left(x_{1},x_{2},\cdots ,x_{T}\right)$, the LSTM computes the hidden vector sequence $h=\left(h_{1},h_{2},\cdots ,h_{T}\right)$ and the output predicted sequence $y=\left(y_{1},y_{2},\cdots ,y_{T}\right)$ by iterating the following equations:(4)$$h_{t}=H(W_{xh}x_{t}+W_{hh}h_{t-1}+b_{h})$$
(5)$$y_{t}=W_{hy}h_{t}+b_{y}$$
where the *W* term denotes weight matrices (e.g., $W_{xh}$ is the input-hidden weight matrix), the b term denotes bias vectors (e.g., $b_{h}$ is hidden bias vector) and H is the hidden layer function.

For more information about the LSTM model, please refer to Reference [30]. In the model of LSTM, there are integer hyper-parameter num_units and batch_size that affect the performance. In the RMSprop optimization, there are continuous hyper-parameters, lr and rho, which also affect the performance of LSTM. Thus, there are both integer and continuous hyper-parameters affecting the performance of LSTM.

## 4. Validation

In this section, we validate the proposed method that used the cuckoo algorithm and the improved algorithm to solve the hyper-parameters problems in LS-SVM and LSTM with the activity recognition datasets in smart home. In doing so, we first optimized hyper-parameters in LS-SVM, based on the basic cuckoo algorithm of Section 4.1. Then, the improved cuckoo algorithms were used to optimize the hyper-parameters in LSTM of Section 4.2.

### 4.1. Hyper-Parameter Optimization in LS-SVM

This subsection verified the effectiveness of the cuckoo algorithm in optimizing hyper-parameters in LS-SVM based on WiFi-based indoor positioning dataset.

Wi-Fi based indoor positioning mainly uses the signal strength value generated by Wi-Fi, that is, the RSSI value, to locate the user’s location. Each hotspot that releases Wi-Fi signals in space is called AP (access point, the signal sent by the wireless router), and one or more APs may be detected at each location. They can be detected at the same time. The AP’s BSSID (that is, the MAC address, which uniquely identifies this hotspot) and the LEVEL value of the Wi-Fi signal (the strength value of the received Wi-Fi signal, in dBm, also known as the RSSI value, Received Signal Strength Indication) need to be obtained The RSSI is continuously collected through a smart phone. First, it is necessary to write an APP and, then, call the Wi-Fi module that comes with the phone to collect and process the data. After that, the phone is saved or sent to the computer for processing.

Due to the instability and fading of the signal in space, the RSSI values measured at the same place and at different times will fluctuate to a certain extent. Therefore, when position matching is performed, corresponding algorithms are required to calculate the distribution of values, and then test this to predict the distribution of values. For example, input the corresponding coordinate matrix and RSSI matrix for training data, and then input the tested RSSI vector to predict the coordinate value to complete the positioning.

Here we used the LS-SVM to predict the user’s location based on the RSSI value generated by Wi-Fi. Since the model parameter gam and sigma of the least square support vector machine have a great influence on the position prediction effect, we first used the cuckoo search algorithm to optimize the optimal gam and sigma, and then used the least square support vector machine to determine the position so as to make an estimate.

Errors continuously exist. Assuming that the real coordinates are (*x*0, *y*0) and the predicted coordinates are (*x*, *y*), the error between the predicted position coordinates and the real position coordinates in this paper was defined as:(6)$$\mathrm{error}=\sqrt{{\left(x0-x\right)}^{2}+{\left(y0-y\right)}^{2}}$$

Before optimization, we used the empirical parameters gam = 0.001, sigma = 0.05 as the initial points, and the search intervals of gam and sigma were both set to [0.001, 1000]. The predictions errors of the training position with iterate 10, 100 and 1000 times respectively are shown in Figure 2.

Since the optimized parameters of iterate 1000 have the lowest prediction error for the training position, we used this parameter to predict and estimate 33 test positions during the test. Figure 3 is a schematic diagram of the errors before and after prediction at 33 locations, o is the estimated error of the empirical parameter gam = 0.001, sigma = 0.05 for each test location, * is the estimated error of the optimized parameter gam = 0.001, sigma = 4.3507 for each test location. From the figure, we could see that the optimized parameters greatly reduced the estimation error of the test data.

Since the coordinates of the 33 tested locations were very close to each other, we only gave a schematic diagram of the prediction comparison of the first 8 locations. Figure 4 is a schematic diagram of the comparison of predictions before and after optimization. In the figure, the *i*-th real coordinate position is marked as + with (L,1),…(L,8). The figure representing the predicted position of the *i*-th coordinate using the parameters before optimization (gam = 0.001, sigma = 0.05) is marked as* with (1,p1),…(8,p1). The predicted position of the i-th coordinate using optimized parameters (gam = 0.001, sigma = 4.3507) is marked as o with (1,p2),…(8,p2). From the figure, it can be seen that before optimization, only 2 positions could be accurately positioned and after optimization, all 8 positions could be accurately positioned. Thus, after the cuckoo optimized the hyper-parameters, the LS-SVM model predicted the positions more accurately.

### 4.2. Hyper-Parameter Optimization in LSTM

This subsection validated hyper-parameter optimization in LSTM with the ADL Adlnormal dataset [31] and Kasteren Dataset [32]. Adlnormal dataset was collected in a smart apartment test bed located on the WSU campus. The dataset recorded 24 WSU undergraduate students performing five ADLs, one at a time. For the 6425 samples of the data set, we divided them into three parts: 3000 for the training data set, 2000 for the validation data set, and 1425 for the test data set. The Kasteren dataset was collected in a three-room apartment where a 26-year-old man lives and there were 14 state-change sensors installed in this apartment. We conducted experiments using the previous 30,000 time slices and split the training, validation and test datasets into three equal parts, i.e., 10,000 time slices, respectively.

In the model of LSTM, there are integer hyper-parameters num_units and batch_size that affect the performance. In the RMSprop optimization, there are continuous hyper-parameters, lr and rho, which also affect the performance of LSTM. Thus, there are both integer and continuous hyper-parameters affecting the performance of LSTM. This subsection used the proposed hyper-parameter optimization algorithm to optimize continuous hyper-parameters, integer hyper-parameters and mixed hyper-parameters, respectively.

In order to verify the effectiveness of our proposed method, the activity recognition accuracy obtained by this method was not only compared with the empirical value, but also compared with the activity recognition accuracy obtained by Optuna [25] optimizing hyper-parameters. Finally, this subsection compared the accuracy of LSTM activity recognition under different hyper-parameter optimization strategies, and analyzed the impact of different strategies on the model.

#### 4.2.1. Experiment 1

This experiment validated continuous hyper-parameter optimization in LSTM. To optimize the two continuous hyper-parameters, lr and rho, we set other hyper-parameters to constants (num_units = 128, batch_size = 200). The initial lr and rho were initialized as [0.001, 0.9], which is the default value of RMSprop, and the search intervals for hyper-parameters were lr∈[0.001,0.01], rho∈(0.1,0.99).

After CS optimization, the continuous hyper-parameters for the Adlnormal dataset were lr = 0.00782101 and rho = 0.59629055, and the continuous hyper-parameters for the Kasteren Dataset were lr = 0.00381946, rho = 0.56684786. It could be seen that the optimized continuous hyper-parameters of LSTM were different for the different data sets. 

Before continuous hyper-parameter optimization, the accuracy score of test data set for Adlnormal dataset was 0.7986 and the accuracy score of test data set for Kasteren Dataset was 0.8445. After continuous hyper-parameter optimization with CS, the accuracy score of test data set for Adlnormal dataset was 0.8529 and the accuracy score of test data set for Kasteren Dataset was 0.8537. Figure 5 compares the accuracy score of the test data set with initialized hyper-parameters, hyper-parameter optimization with CS and hyper-parameter optimization with Optuna, where CHO represents only continuous hyper-parameters, lr and rho, optimized. From the figure, we can see that the activity recognition accuracies of the two datasets both improved significantly after continuous hyper-parameter optimization with CS and CS was better than Optuna.

#### 4.2.2. Experiment 2

This experiment validated integer hyper-parameter num_units and batch_size optimization in LSTM, based on the improved Cuckoo optimization Algorithm 1 with the ADL Adlnormal dataset and Kasteren dataset. 

To optimize the integer hyper-parameter num_units and batch_size, we set continuous hyper-parameters to constants (lr = 0.001, rho = 0.9). The integer hyper-parameters were initialized as num_units = 128, batch_size = 200, and we set the search interval of num_units and batch_size to [1, 256] and [1, 1000], respectively.

Set epochs = 10, the integer hyper-parameters after optimization for Adlnormal dataset were num_units = 253, batch_size = 491, and the integer hyper-parameters after optimization for Kasteren Dataset were num_units = 12, batch_size = 931. It could be seen that the optimized integer hyper-parameters of LSTM were different for the different data sets.

After integer hyper-parameter optimization, the accuracy score of test data set for Adlnormal dataset was 0.8220 and the accuracy score of test data set for Kasteren Dataset was 0.8560. Figure 6 compares the accuracy score of the test data set with initialized hyper-parameters, hyper-parameter optimization with CS and hyper-parameter optimization with Optuna, where IHO represents only integer hyper-parameter num_units and batch_size optimized. From the figure, we can see that the activity recognition accuracies of the two datasets both improved significantly after integer hyper-parameter optimization with CS and CS was better than Optuna.

#### 4.2.3. Experiment 3

This experiment validated mixed hyper-parameters lr, rho, num_units and batch_size optimization in LSTM, based on the improved Cuckoo optimization Algorithm 2 with the ADL Adlnormal dataset and Kasteren dataset.

To optimize the continuous and integer hyper-parameters together, the mixed hyper-parameters were initialized as lr = 0.001, rho = 0.9, num_units = 128, batch_size = 200, and the search intervals for hyper-parameters were set as continuous hyper-parameters and integer hyper-parameters above.

Set epochs = 10, the mixed hyper-parameters after optimization for Adlnormal dataset were lr = 0.00989974980, rho = 0.765867432, num_units = 8, batch_size = 78, and the integer hyper-parameters after optimization for Kasteren Dataset were lr = 0.00793324624, rho = 0.758825652, num_units = 129, batch_size = 129. It could be seen that the optimized mixed hyper-parameters of LSTM were different for the different data sets. 

After mixed hyper-parameters optimization, the accuracy score of test data set for Adlnormal dataset was 0.8446 and the accuracy score of test data set for the Kasteren dataset was 0.8693. Figure 7 compares the accuracy score of the test data set with initialized hyper-parameters, hyper-parameters optimization with CS and hyper-parameters optimization with Optuna, where MHO represent optimized mixed hyper-parameters lr, rho, num_units and batch_size together with lr = 0.001, rho = 0.9, num_units = 128 and batch_size = 200 as the initial value of optimization. From the figure, we can see that the activity recognition accuracies of the two datasets both improved significantly after mixed hyper-parameter optimization with CS and CS was better than Optuna. 

#### 4.2.4. Experiment 4

This experiment compared the accuracy of LSTM activity recognition under different hyper-parameter optimization strategies, and analyzed the impact of different strategies on the model with the ADL Adlnormal dataset and Kasteren Dataset.

The optimized hyper-parameter and activity recognition accuracy with different strategies are shown in Table 1, where CHO and IHO meant optimizing continuous hyper-parameters, lr and rho, and optimizing integer hyper-parameters, num_units and batch_size, separately, and finally, the separately trained parameters were merged together for activity recognition. CHO after IHO meant optimizing continuous hyper-parameters, lr and rho, with optimized integer hyper-parameters, num_units and batch_size, as input. IHO after CHO meant optimizing integer hyper-parameters, num_units and batch_size, with optimized continuous hyper-parameters, lr and rho, as input. MHO after CHO and IHO meant optimizing mixed hyper-parameters with CHO and IHO result as the initial value of optimization.

Figure 8 and Figure 9 compared the accuracy score of different hyper-parameter optimization strategies for Adlnormal dataset and Kasteren Dataset, respectively. From the figure, we can see that the activity recognition accuracy was improved after hyper-parameter optimization for all optimization strategies. Compared with integer parameters, continuous parameters had a greater impact on the LSTM. Mixed hyper-parameter optimization obtained a stable improvement effect. The optimization strategies CHO and IHO, CHO after IHO, IHO after CHO, MHO after CHO and IHO also obtained relatively good effects. 

## 5. Conclusions

In order to reduce the hyper-parameter settings influence of certain algorithms on activity recognition, this paper proposed to use the cuckoo optimization algorithm to optimize the hyper-parameters in the algorithms, and improved the cuckoo algorithm to adapt to the optimization of integer hyper-parameters and hybrid hyper-parameters in activity recognition problems.

To validate the proposed method, this paper first optimized the hyper-parameters in LS-SVM, based on the basic cuckoo algorithm, and compared the WiFi-based indoor localization results before and after optimizing the hyper-parameters. The experimental results showed that after CS optimized the hyper-parameters, the LS-SVM model predicted the positions more accurately. Then, this paper validated hyper-parameter optimization in LSTM with the ADL Adlnormal dataset and Kasteren Dataset and compared the activity recognition accuracy of CS-optimized hyper-parameters, empirical hyper-parameters, and Optuna optimized hyper-parameters. Experimental results showed that the optimized hyper-parameters of LSTM were different for different data sets and optimizing hyper-parameters with CS obtained the best activity recognition accuracy compared with empirical hyper-parameters, and Optuna optimized hyper-parameters. Finally, this paper compared the accuracy of LSTM activity recognition under different hyper-parameter optimization strategies, and analyzed the impact of different strategies on the model. The result showed that the mixed hyper-parameter optimized with the improved cuckoo algorithm obtained a stable improvement effect. The other optimization strategies also obtained relatively good effects.

Each contribution to activity recognition brings us one step closer to the realization of Ambient Intelligence. As future work, we plan to expand hyper-parameter optimization in LS-SVM and LSTM to other activity recognition algorithms, and expand the cuckoo algorithm hyper-parameter optimization to other artificial intelligence algorithms, such as particle swarm and wolf swarm algorithms.

## Figures and Tables

**Figure 1 entropy-24-00845-f001:**
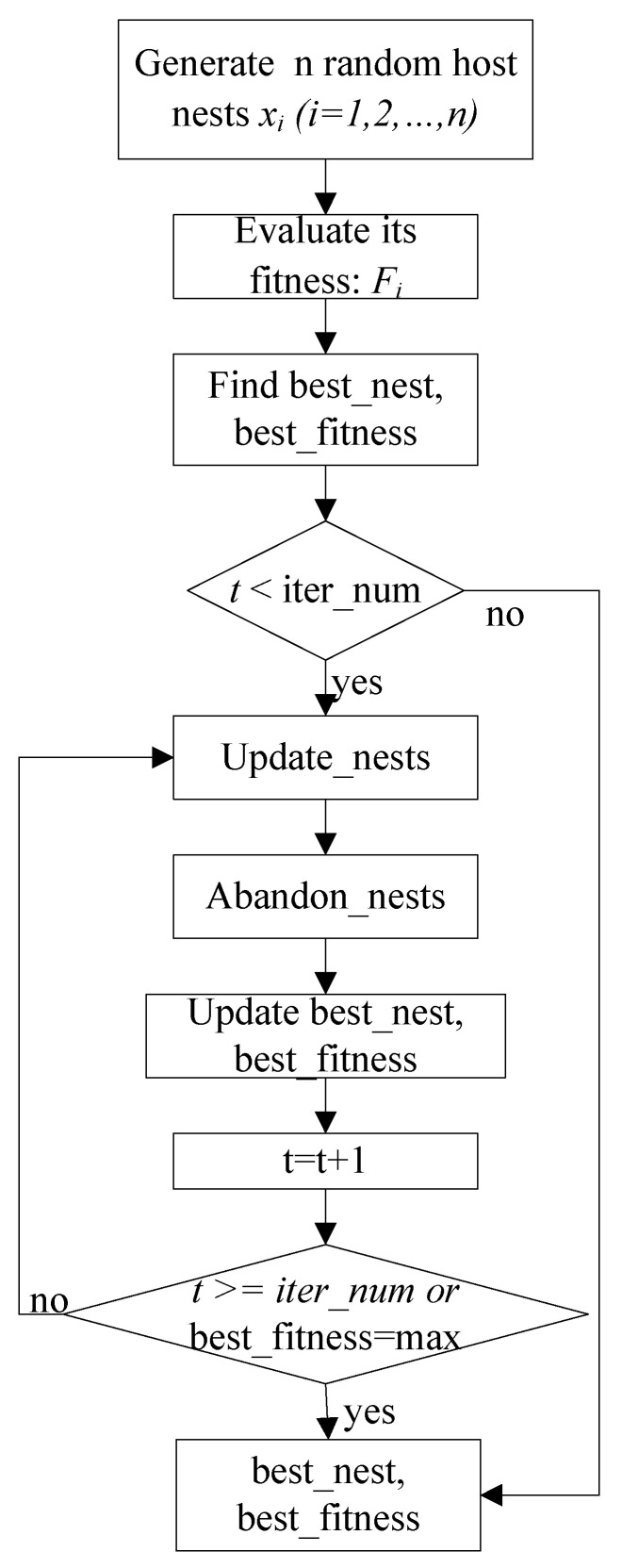
The traditional CS algorithm flowchart.

**Figure 2 entropy-24-00845-f002:**
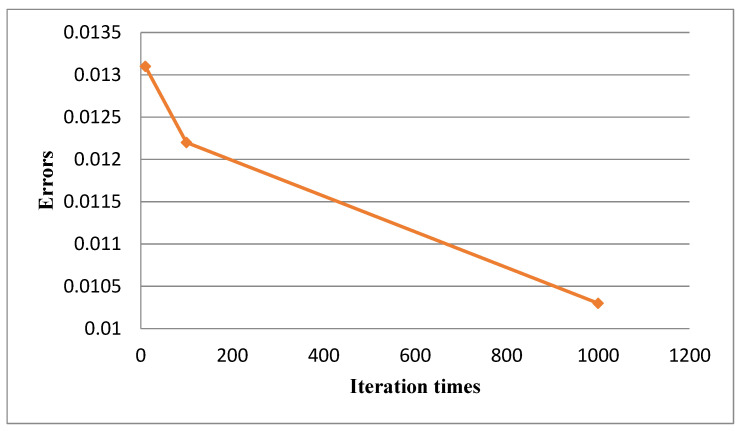
The prediction errors of the training position with iterates 10, 100 and 1000 times, respectively.

**Figure 3 entropy-24-00845-f003:**
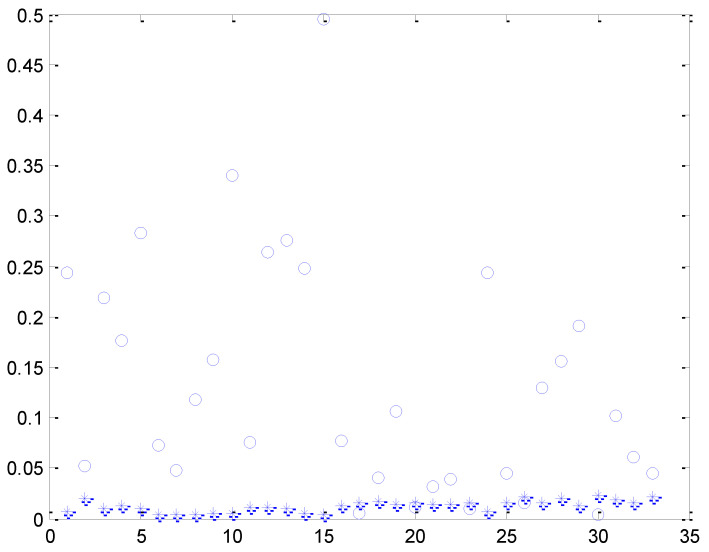
Prediction errors at 33 positions before and after optimization, (o) the estimated error of the empirical parameter, (*) the estimated error of the optimized parameter.

**Figure 4 entropy-24-00845-f004:**
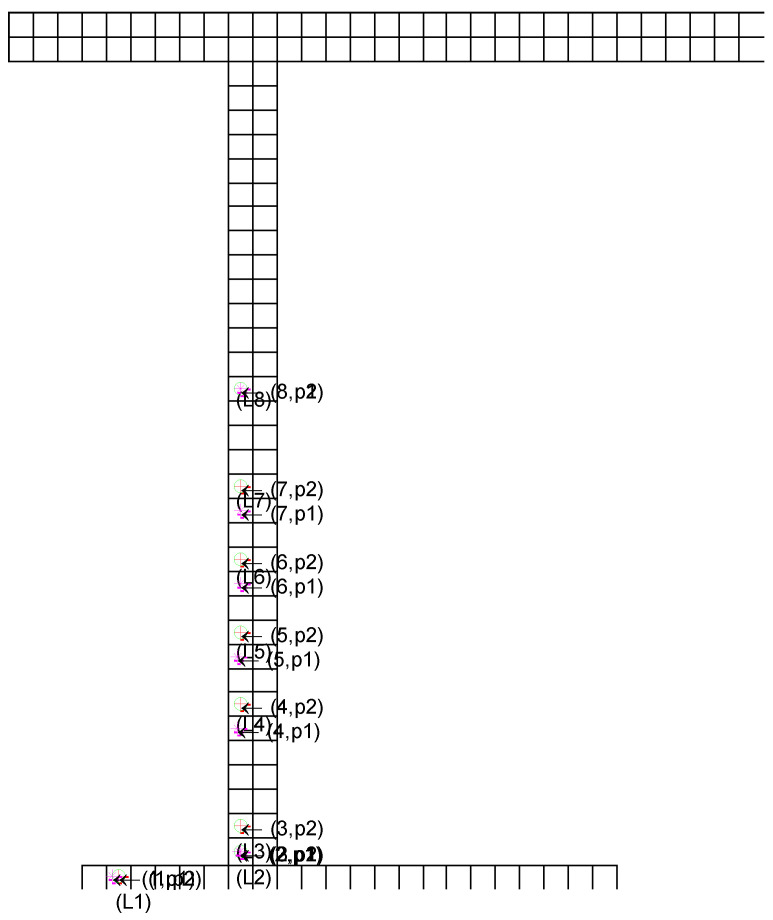
The schematic diagram of real coordinate position (+), prediction positions without CS optimization (*) and prediction positions with CS optimization (o).

**Figure 5 entropy-24-00845-f005:**
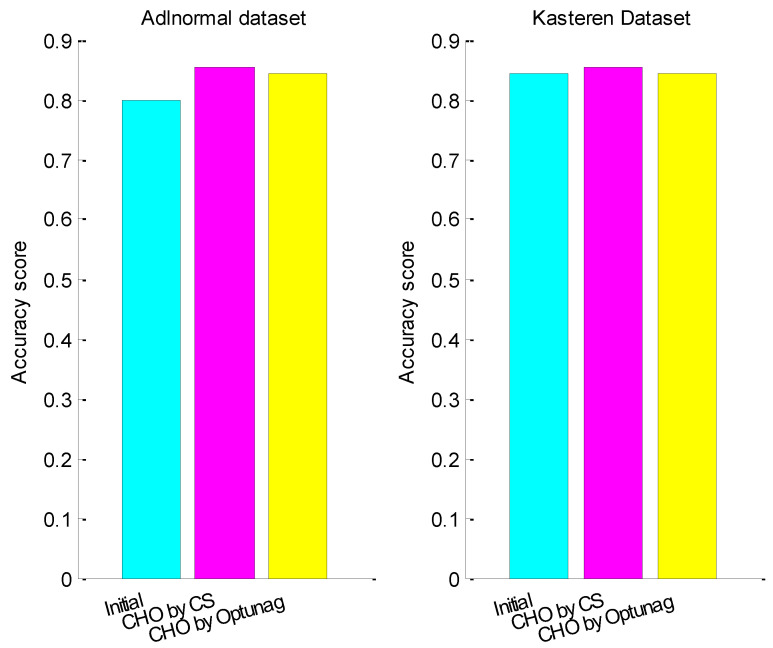
The accuracy score before and after continuous hyper-parameters optimization.

**Figure 6 entropy-24-00845-f006:**
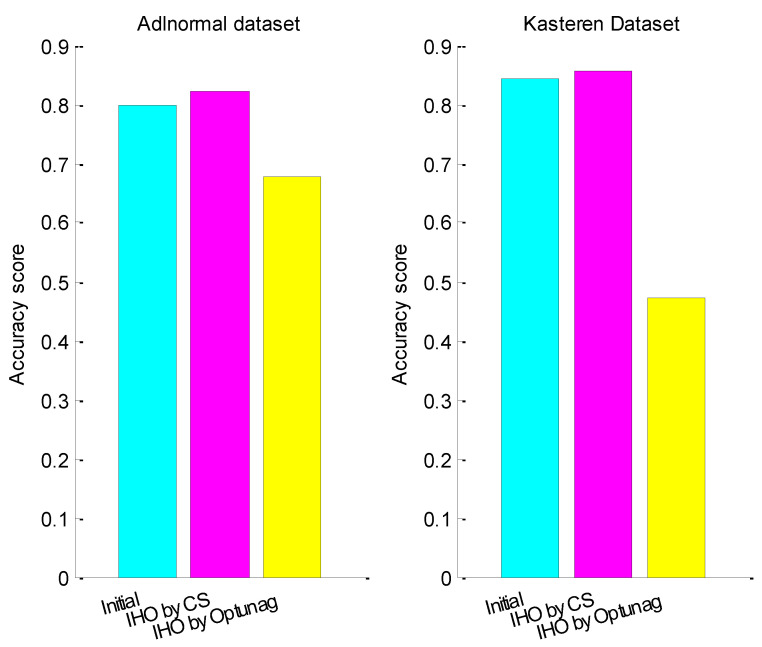
The accuracy score before and after integer hyper-parameters optimization.

**Figure 7 entropy-24-00845-f007:**
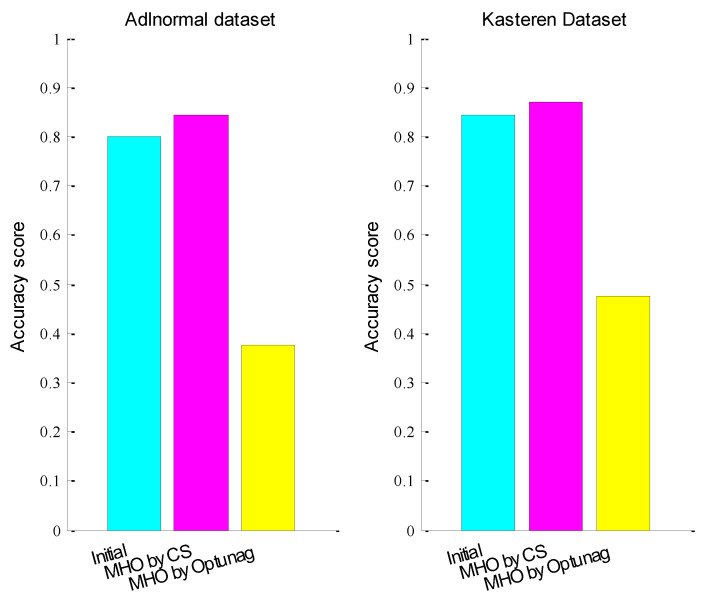
The accuracy score before and after mixed hyper-parameters optimization.

**Figure 8 entropy-24-00845-f008:**
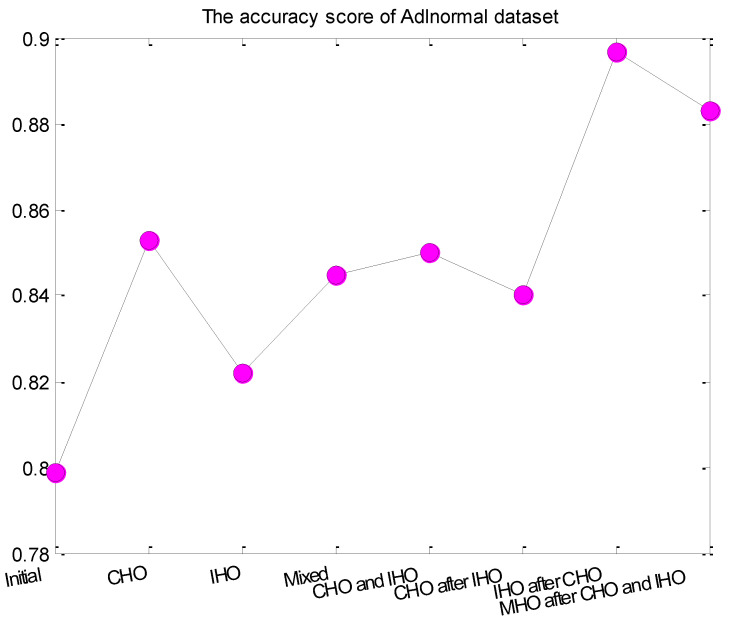
The activity recognition accuracy of different hyper-parameter optimization strategies for Adlnormal dataset.

**Figure 9 entropy-24-00845-f009:**
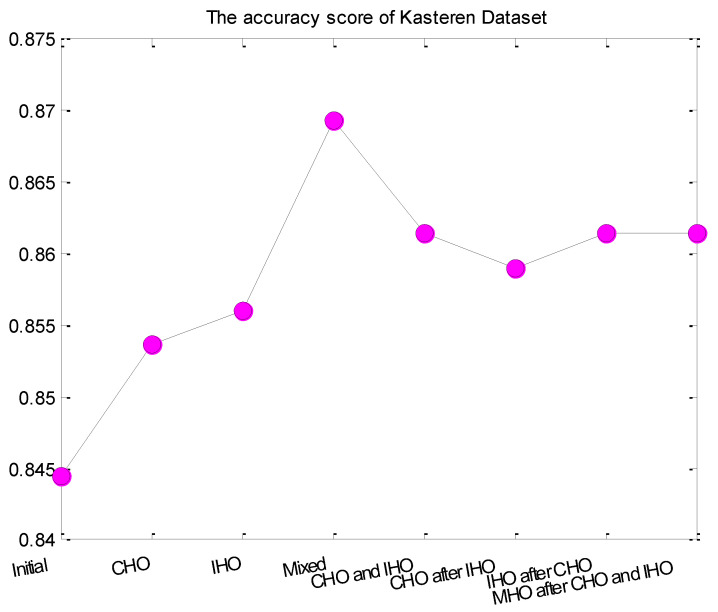
The activity recognition accuracy of different hyper-parameter optimization strategies for Kasteren Dataset.

**Table 1 entropy-24-00845-t001:** The optimized hyper-parameter with different strategies.

	Hyper-Parameters of Adlnormal Dataset	Hyper-Parameters of Kasteren Dataset
**Initial hyper-parameters**	(0.001, 0.9, 128, 200)	(0.001, 0.9, 128, 200)
**CHO**	(0.00782101, 0.59629055, 128, 200)	(0.00381946, 0.56684786, 128, 200)
**IHO**	(0.001, 0.9, 253, 491)	(0.001, 0.9, 12, 931)
**Mixed**	(0.00989974980, 0.765867432, 8, 78)	(0.00793324624, 0.758825652, 129, 129)
**CHO and IHO**	(0.00782101, 0.59629055, 253, 491)	(0.00381946, 0.56684786, 12, 931)
**CHO after IHO**	(0.00528674, 0.72591224, 253, 491)	(0.0095465, 0.78940525, 12, 931)
**IHO after CHO**	(0.00782101, 0.59629055, 187, 1)	(0.00381946, 0.56684786, 141, 73)
**MHO after CHO and IHO**	(0.00934384542, 0.634805436, 1, 64)	(0.00501521055, 0.97690847, 77, 44)

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
