# Peer review of "Research on Hyper-Parameter Optimization of Activity Recognition Algorithm Based on Improved Cuckoo Search"

_entropy, 2022, doi:10.3390/e24060845_

Round 1

Reviewer 1 Report

I am satisfied with the responses provided by the authors. Still, minor grammatical mistakes and typos are there. Please check it once again before publication. In few places, references are not cited properly. It is suggested to keep the flowchart of the modified CS (optional). The authors should also discuss why Algorithm 1 is for LS-SVM and Algorithm 2 is for LSTM. Why not the same algorithm for both purposes? Include some discussions on this aspect. 

Reviewer 2 Report

The authors have addressed my major concerns. Therefore, I believe that the paper can be accepted for publication. 

Some minor issues:

- [Error! Bookmark not defined.] in Page 5

- Figure 4 is not readable

This manuscript is a resubmission of an earlier submission. The following is a list of the peer review reports and author responses from that submission.

Round 1

Reviewer 1 Report

The authors propose a variant of the cuckoo algorithm for optimizing continuous and integer hyper-parameters on activity recognition methods. They apply the proposed algorithm to optimize the hyper-parameters in LS-SVM and LSTM. The results show that the new variant of the proposed algorithm can effectively improve the performance of the model in activity recognition.

The paper will of interest to the readers of this journal since the application area of the new variant of the proposed area is important to the field. However, I believe that there are some issues that need to be addressed by the authors before publication.

Major issues:
- The literature review is incomplete since some important methods for hyper-parameter tuning are not mentioned at all. More specifically, researchers have utlized derivative-free methods for parameter tuning and they handled both continuous and integer variables, e.g.:

1. Sauk, B., Ploskas, N., & Sahinidis, N. (2020). GPU parameter tuning for tall and skinny dense linear least squares problems. Optimization Methods and Software, 35(3), 638-660.
2. Liu, J., Ploskas, N., & Sahinidis, N. V. (2019). Tuning BARON using derivative-free optimization algorithms. Journal of Global Optimization, 74(4), 611-637.
3. Koch, P., Golovidov, O., Gardner, S., Wujek, B., Griffin, J., & Xu, Y. (2018, July). Autotune: A derivative-free optimization framework for hyperparameter tuning. In Proceedings of the 24th ACM SIGKDD International Conference on Knowledge Discovery & Data Mining (pp. 443-452).

- The authors do not compare their results with a state-of-the-art hyperparameter tuner like Optuna. Doing so will add much added value to the paper.

Minor issues:

- Please provide the definition before the abbreviation, e.g., Hidden Markov Models for HMM
- When an abbreviation is defined, please use this abbreviation throughout the text
- The numbers of Sections "Validation" in Page 5 and "Conclusions" in Page 12 are incorrect.
- The are various language issues/typos, e.g., "a typical traditional methods" in Page 1, "will validate optimize" in Page 5

Reviewer 2 Report

The novelty of the paper is very poor. The paper requires correction in all aspects. Improve the paper quality and submit it elsewhere.